# A Review of Probe-Based Enrichment Methods to Inform Plant Virus Diagnostics

**DOI:** 10.3390/ijms25158348

**Published:** 2024-07-30

**Authors:** Thomas Farrall, Jeremy Brawner, Adrian Dinsdale, Monica Kehoe

**Affiliations:** 1Plant Innovation Centre, Australian Government, Department of Agriculture, Fisheries and Forestry (DAFF), Canberra, ACT 2601, Australia; thomas.farrall@research.usc.edu.au (T.F.); adrian.dinsdale@aff.gov.au (A.D.); 2Forest Research Institute, School of Science, Technology and Engineering, University of the Sunshine Coast, Sippy Downs, QLD 4556, Australia; 3Plant Pathology Department, University of Florida, Gainesville, FL 32611, USA; 4Diagnostic Laboratory Services, Biosecurity and Sustainability, Department of Primary Industries and Regional Development (DPIRD), Perth, WA 6151, Australia

**Keywords:** hybridisation capture, next-generation sequencing, viruses

## Abstract

Modern diagnostic techniques based on DNA sequence similarity are currently the gold standard for the detection of existing and emerging pathogens. Whilst individual assays are inexpensive to use, assay development is costly and carries risks of not being sensitive or specific enough to capture an increasingly diverse range of targets. Sequencing can provide the entire nucleic acid content of a sample and may be used to identify all pathogens present in the sample when the depth of coverage is sufficient. Targeted enrichment techniques have been used to increase sequence coverage and improve the sensitivity of detection within virus samples, specifically, to capture sequences for a range of different viruses or increase the number of reads from low-titre virus infections. Vertebrate viruses have been well characterised using in-solution hybridisation capture to target diverse virus families. The use of probes for genotyping and strain identification has been limited in plants, and uncertainty around sensitivity is an impediment to the development of a large-scale virus panel to use within regulatory settings and diagnostic pipelines. This review aims to compare significant studies that have used targeted enrichment of viruses to identify approaches to probe design and potential for use in plant virus detection and characterisation.

## 1. Introduction

Advancements in high throughput sequencing (HTS) technologies have greatly improved the potential for the detection, identification, and characterisation of pathogens. Industries such as biosecurity traditionally employ a mix of conventional pathology and molecular diagnostic techniques for the detection of pathogens on plants and plant products. Nucleic acid amplification methods such as polymerase chain reaction (PCR) [1], quantitative PCR (qPCR) [2], and antibody-based methods such as an enzyme-linked immunosorbent assay (ELISA) [3] have provided reliable assays for pathogen detection in regulatory settings. However, as the importation of high-risk plant commodities increases, traditional diagnostic methods are at risk of failing to keep up with increasing demand and changes in pathogen populations. As the list of emerging pathogens expands, so does the list of new assays that must be developed and validated for implementation [4]. Inhibitors that can be introduced into nucleic acid testing workflows from large sampling sizes can reduce the effectiveness and sensitivity of such tests. High-throughput sequencing or next-generation sequencing (NGS) allows for large quantities of DNA and RNA to be sequenced in parallel to identify multiple pathogens in a single sample. Nucleic acid sequencing technologies have the potential to address the concerns associated with the growing list of pathogens that require detection and identification. 

The HTS platforms developed by Pacific Biosciences (Menlo Park, CA, USA) (PacBio) and Oxford Nanopore Technologies (Oxford, UK) (ONT) are third-generation sequencing (TGS) systems that provide extremely long reads by sequencing individual nucleic acid molecules [5,6]. ONT sequencing has been used to produce thousands of full-length, high-quality draft virus genome sequences, which were not recovered using standard short-read assembly approaches [7]. The ONT system is of particular interest for disease diagnostics due to the large volumes of data that are produced, portability, the speed of setup, and the ability to deploy relatively low-cost and portable equipment in traditional diagnostic laboratories [8]. While effective for the detection of diseases, limitations include the basecalling accuracy, notably present when sequencing homopolymers and modifications [6]. These error rates in basecalling can be attributed to the basecallers themselves and associated algorithms rather than a product of the sequencing platform’s capabilities [6]. Increases in the availability of public databases and the abundance of genetic information therein have been particularly useful for the taxonomic classification of pathogens and hosts [9,10].

However, significant validation is required to ensure confidence in HTS as a routine diagnostic tool, particularly in highly regulated biosecurity settings. The sensitivity of sequencing-based workflows to detect target pathogens needs to be validated before being considered a viable detection tool. Concerns arise from whether sequencing alone is sensitive enough to detect all pathogens, including novel disease-causing agents. Targeted HTS using probes that hybridise with specific DNA sequences and are subsequently captured for sequencing [11], hybridisation capture (HybCap) has provided ways to focus resources on selected highly informative regions of the genome. Enriching samples using probes can increase the sensitivity of target regions prior to sequencing. Probes, also referred to as ‘baits’ in some literature, may be designed to target regions of pathogen genomes so they may be extracted from nucleic acids for sequencing and subsequent pathogen identification. HybCap has been demonstrated in vertebrate species and selected bacterial and fungal pathogens in plants [12,13,14]. The technique has also been used to create highly resolved phylogenies describing similarities among organisms and to identify organisms [13,15,16,17,18,19,20,21]. While the methods are well developed for double-stranded DNA (dsDNA) viruses in humans [22], questions remain as to whether a plant virus probe panel targeting single-stranded RNA viruses (ssRNA) or their cDNA complements may be designed to overcome the shortcomings associated with sequencing plant viral material; specifically, adequate sensitivity, inhibitor mitigation, and cost-effectiveness. 

Many problems associated with the molecular diagnostics of plant pathogens arise from compounds present in host tissue that inhibit enzymes used in molecular diagnostic assays. The composition of seed tissue, in particular, makes the extraction of nucleic acids to produce high-quality templates particularly challenging, yet testing is highly relevant for the management of risks from seed-borne and seed-transmitted pathogens [23,24,25]. 

This review aims to compare probe-based hybridisation for capture and sequencing in order to identify pathogens. Specific emphasis has been placed on how these methods can overcome the challenges associated with virus targets in vertebrates and the use of probes to characterise bacterial and fungal strains in plant host tissue. 

## 2. Notable Advancements in Target Enrichment

### 2.1. HTS in Plant Diagnostics

To accommodate the growing demand for pathogen testing in plant commodities, there has been a large push towards nucleic acid-based testing and large-scale parallel sequencing in the form of metagenomics [26]. HTS workflows are advantageous as they do not require knowledge of specific aspects of viral agents (e.g., species-specific primer binding regions) to obtain sequence data, unlike pathogen-specific nucleic-acid-based assays [27]. 

HTS methodologies involve three general steps: fragmentation of the DNA template, addition of sequencing adapters to the nucleic acid library, and sequencing of each fragment. Depending on the starting material, there may be additional steps for enriching the sample and for the removal of non-target material such as ribosomal RNA (rRNA); rRNA depletion is particularly important for the sequencing of RNA. These fragments can be sequenced to provide enough depth and coverage to allow for pathogen identification, assuming a reference sequence is available in the diagnostician’s database. HTS workflows have been compared extensively [28,29,30] and, therefore, will not be covered in detail here.

### 2.2. Sample Enrichment

When using HTS for pathogen screening or detection, if the pathogen resides at a low titre in the host, then insufficient sequence data could be obtained to make a detection or identification. Selective targeting prior to sequencing may limit the number of non-target sequences generated prior to the sequencing step, increasing overall assay sensitivity. This can be performed by enriching for targets or by rejecting non-target material. 

By isolating specific regions, informative sequences can be captured using targeted approaches, and these enriched samples can then be sequenced with HTS methods. This can increase the sensitivity of pathogen detection and provide higher sequencing resolution, aid the discovery of novel pathogens, and allow for the use of high-throughput techniques to capture data from many sequences in parallel. 

There are numerous approaches and diverse methods classified under the enrichment umbrella. Hybridisation capture, PCR/amplicon enrichment, and background depletion are all ways in which primers or probes can be applied to capture a target [31] (Table 1). 

Target enrichment methods have been used previously for the identification and characterisation of viruses, bacteria, and fungi in vertebrate hosts [32]. HTS libraries are fragmented and then hybridised into specific probes to capture the region of interest. Sequencing is focused on the regions that are enriched to improve the efficiency of sequencing in a similar way that in-silico enrichment is used to obtain an analysis of data that is less computationally intensive when compared to metagenomic approaches. 

The Rainstorm platform is one of the earliest uses of droplet PCR enrichment, which involves droplets containing primer pairs that are subject to thermal cycling and shotgun sequencing to recover many target regions in parallel. Rainstorm overcomes the main shortcomings associated with multiplex PCR, in that primer pairs cannot compete against each other or interact, reducing the chances of primer dimers and increasing uniformity for selected targets. However, this is limited to a maximum of a 2–3 Mb target size, making it unsuitable to target larger genomes [33]. However, the reliance on PCR means the possible introduction of PCR bias, such as DNA recombination, prior to sequencing. 

Modern enrichment methods utilise newer technologies, such as the pore-based sequencing offered by ONT. The pores allow for ‘adaptive’ sampling, which means that nucleic acid sequences can be ejected from a sequencing pore in real-time. This can reduce the number of off-target reads but at the cost of increased pore degradation caused by the rejection of sequences [34]. Another modern enrichment technique is the use of CRISPR-based enzymes. These ‘molecular scissors’ can be used to cut and isolate target regions while also hybridising guides to the target [35] (Figure 1). These are powerful tools, but like many enrichment methods, their efficiency is dependent on the sample and conditions in which they are used. 

### 2.3. Hybridisation Capture

There are many techniques that utilise hybridisation capture specifically, which often have been referred to as hybrid capture, target capture, gene capture, in-solution hybridisation, and capture-based probe hybridisation [36]. These all describe methods in which specific probe sequences are hybridised to specific regions of a genome from a host or pathogen so that these sequences can be isolated to enhance sensitivity. The depth of sequencing that is possible with HybCap-based methods can provide more accurate estimates of copy number and improve the detection of structural rearrangements [37]. 

HybCap was first used to capture and sequence the human exome using an array-based method in 2007 [38]. This approach has been used to identify variants and mutations that may cause disease; however, a limitation of this approach was the need to design probes for the capture of a large target region. The array-based method has since been optimised to an in-solution method that overcomes some of the sensitivity and specificity issues associated with on-array HybCap and also requires less starting DNA material [31]. Gene capture in the form of arrays can also detect divergent sequences even if probes are based on individual reference genomes [36,39]. Gnirke et al. (2009) used a considerably lower amount of 170 bp RNA probes to capture regions of the human exome and overcome some of the shortcomings associated with using 60–90 bp DNA probes that were used in Nimblegen arrays from previous methods [36,38,40].

Since the enrichment of the human exome, HybCap has been demonstrated in human hosts for pathogen enrichment. It has been used in human hosts to enrich low-titre unicellular parasite Plasmodium falciparum sequences at a 37–44-fold enrichment [36,41]. Additionally, this method has also been used to target bacterial pathogens, specifically Borrelia burgdorferi in humans, which was successfully captured with 66.45% efficiency and 99.5% coverage [42]. Despite successes in vertebrae, successful capture can vary and depends on compounding factors. The workflow and design of the probes are both equally important in achieving successful capture. 

## 3. Virus Enrichment and Capture

The detection of viruses using HybCap presents a large set of challenges. Unlike bacteria or fungi genomes, which are significantly larger, virus genomes are comparatively shorter in length, thus limiting the amount of sequence available for probe design. There is also a significant lack of conserved genes between viruses; with no gene shared among all virus families, there is no universal amplicon-based sequencing approach. Studies involving larger genome targets such as bacteria and fungi have many conserved regions such as ITS, 16s, and 28s; virus sequence variation presents a unique challenge in the lack of conserved genes, further emphasising the need for targeted probe design. Metagenomic shotgun sequencing of all nucleic acids in samples has been used to identify virus samples and is a highly inefficient use of sequencing resources as virus reads from metagenomic sequencing are extremely rare compared to reads from plant and animal hosts [32]. Assessing the sequence content in clinical virus samples has proven to be difficult due to viruses being largely understudied. High sequence diversity in virus groups also compounds these issues [43]. Studies targeting human viruses have used a variety of methods for probe selection in attempts to capture this diversity [32,44,45,46]. Along with probe design, library preparation and enrichment workflows influence the capture of viruses seen in vertebrate hosts. 

### 3.1. Library Preparation and Enrichment Types 

The role of library preparation before sequencing, post-capture prior to sequencing, and the effect of fragmentation are important to consider. Longer fragments may lead to a higher rate of off-target reads, whereas shorter fragments may lead to a higher specificity of capture [47]. However, longer reads do lead to an increase in the confidence of alignment and provide an increase in specificity. 

There are many variations in library preparation that are suitable for different purposes: transposon-mediated fragmentation or tagmentation, where transposomes are used to fragment DNA and add adapters in a single step, which decreases hands-on time [47,48]. The tagmentation method of enrichment has been seen to enrich human viruses for sequencing of libraries using TaME-seq [49,50] (Table 1). Traditionally, TaME-seq has been used as an alternative to capture panels due to the increased cost of capture panels; however, they can be used in tandem. Aside from tagmentation, various kinds of enrichment methods have been developed, such as molecular inversion probes and the Haloplex method (biotin-labelled circularisation probes bind to endonuclease-digested DNA followed by extension ligation) [51]. This method involves universal primers that bind to the circular DNA for PCR amplification and has been demonstrated in human oncology [47,51,52]. Hook probes or hook-ligation can also be used in the hook-capture method, which uses the CircLigase enzyme to ligate single-stranded DNA or RNA in human hosts [47,53]. Universal primers form dsDNA, and the remaining single strands are digested. This process removes the need for streptavidin and biotin beads, making it more affordable and simplifying library preparation. PolyA-based enrichment has been used for RNA and DNA enrichment, but only for viruses with a polyA tail [54,55]. While these methods are effective, enrichment for detection and identification in complex background tissue needs to be targeted, and the use of universal primers prior to sequencing may result in less targeted enrichment [56]. 

Singh (2022) found that DNA sonication leads to better library uniformity than digestion methods. HybCap-based methods such as SureSelect and SeqCap EZ performed better across the board than amplicon-based sequencing methods like AmpliSeq, with amplicon-based methods showing more sequencing drop-offs and other sequencing artefacts [47]. 

It is possible to use multiple rounds of enrichment to increase accuracy or yield. Li et al. (2013) [39] targeted single-copy protein-coding genes in gnathostome vertebrate species using two rounds of target capture. This increased enrichment despite using fewer probes per nucleotide target; however, it also increased the time taken for hybridisation. Two to forty-eight hours is considered an appropriate timeframe for most hybridisation studies [47]. By increasing the rounds of target capture, there is an increase in washing steps needed to purify the nucleic acid in the sample.

Increasing washing steps may reduce the amount of viral nucleic acid, but it is necessary to reduce contamination and background sequences. This is tolerable with high-titre pathogens but poses issues with low pathogen numbers. Poor HybCap performance for samples with low-titre or unbiased samples can be common [43]. Using RNA as an input can help to capture more viral content, and small RNA (sRNA) or rRNA-depleted total RNA (totRNA) can be captured. In plant hosts with large amounts of background rRNA, the enrichment of samples is very attractive to proportionally increase the number of viral reads captured [57]. When sequencing for RNA virus identification purposes, direct RNA sequencing is advantageous to avoid any bias that might be introduced through cDNA synthesis and any potential loss of target RNA that may occur through washing and transcribing the large amount of nucleic acid required for sequencing. A limitation is the assembly of sequences using sRNA due to the short read length [55,57]. Whilst library preparation workflow is important, the largest influence on the detection capability of an enrichment workflow is the capacity of capture probes to hybridise with targets, which is based on the design used to create the target panel. 

### 3.2. Impacts of Panel Design on Vertebrate Virus Capture

VirCapSeq-Vert, developed by Briese et al. (2015) [45], is a probe set designed to target all known vertebrate viruses, including those of humans, targeting all known virus taxa with at least one known invertebrate virus. Briese et al. (2015) [45] obtained sequences from the EMBL-coding domain sequence database, clustered at 96% sequence identity, and oligonucleotide sequences diverging by more than 90% were retained to assess sequence variation. For the design consideration of probes, 100 bp oligonucleotide probes were placed with 25–50 bp spacing between them over the target regions. Genomes with as low as 75% sequence identity to probe target sequences have been captured using the VirCapSeq-VERT panel despite not being designed for the discovery of novel viruses. When considering virus detection, the VirCapSeq-VERT panel has the potential to have a capture sequence divergence of 40% when targeting conserved regions [45].

Oligonucleotide probes have also been designed to capture human viruses in clinical samples using the SureSelect (Agilent) target enrichment system for herpesviruses [44,45]. SureSelect has additionally been used to enhance RNA sequencing and has been shown to increase the coverage of bacterial and virus pathogen transcripts when coupled with sequencing [45,58,59]. The Ebola virus and Kaposi’s sarcoma-associated herpesvirus have been enriched using the SureSelect system for characterisation [46,60,61]. This was expanded upon when O’Flaherty et al. (2018) [46] tested a virus panel on respiratory-infecting viruses in humans using two complementary panels (Table 2). The first panel involved virus-specific primers spanning full genomes of ‘common’ respiratory viruses, and the second targeted conserved regions in nine viral families and subfamilies associated with respiratory disease. Although viruses share no conserved genes across families, O’Flaherty et al. (2018) [46] were still able to identify selected conserved regions and motifs across genera and families for selected respiratory viruses. The two complementary libraries used different sequencing indices from the same template to directly compare enriched and non-enriched samples that had their viral load pre-checked using available qPCR assays. In all samples except one, virus-specific probes resulted in an improved read depth compared to the absence of probes. In samples which saw enrichment, the success of enrichment was to varying degrees (increase in target reads for 18 samples was at 50–99%, three samples at 20–45%, and four samples at <10%). The linear genome coverage and depth were improved, and 73% of samples showed more than 85% coverage (Figure 2).

To test the conserved viral probe set, of the 26 successfully enriched viruses, another 27 reference viruses were added from both humans and animals to evaluate the divergence of the probes. Most species were confirmed in the enriched samples and were generally more difficult to identify in the unenriched samples. The conserved group was able to capture all human respiratory viruses as well as the viruses not targeted by the virus-specific probes and viruses that were not enriched using the conserved probes. O’Flaherty et al. (2018) [46] tested mixed samples for two viruses and observed higher reads in one relative to the other. Although genome coverage was not affected, it presented a possible limitation of the HybCap workflow, being that in the event of a co-infection, one enriched virus can saturate the sequence reads and obscure the detection of others. 

Virus-specific probes were the most effective for enrichment where there was greater than 90% similarity with virus sequences. As echoed by Briese et al. (2015) [45], viruses with inconsistent or low sequence homology were more difficult to enrich. The lowest end of this was seen in one clinical sample with no enrichment of the target, which had a 70% similarity with the probe design sequence. Sequencing islands (regions with a high depth scattered along the genome) could be used to fill genome gaps and enhance classification. 

Wylie et al. (2015) [32] targeted 34 virus families using 2.1 million virus and genus-specific probes designed with the NimbleGen SeqCap designer. This included 190 genera and 337 species but excluded human endogenous retroviruses. Nucleic acids were sequenced and compared against a reference sequence for similarity. It saw an increase in the median percentage of viral reads by 674-fold when comparing enriched and non-enriched samples. 

ViroCap was able to demonstrate enrichment in both DNA and RNA and was able to detect divergent sequences from known sequences with as low as 58% sequence similarity. Li et al. (2013) [39] suggest that probes above 60% nucleotide identity promote successful target capture. This is lower than the sequence similarity observed by O’Flaherty et al. (2018) [46]; however, it was not as low of a similarity as the VirCapSeq-VERT panel. This is potentially due to the higher tiling of probes in the later panel, as the ViroCap had an average probe spacing that resulted in an 82 bp gap size compared to 25–50 bp in VirCapSeq-VERT.

Aside from using reference sequences (RefSeq), Wylie et al. (2015) [32] also tiled self-described ‘genome neighbour targets’ where the RefSeq sequences alone may not be enough to capture more divergent strains. This allowed for the detection of sequences that share very little overall sequence similarity to known viruses. However, it still cannot detect sequences that are non-homologous with known sequences. 

### 3.3. Challenges with Evaluating Enrichment Success

This review attempts to identify any distinct trends between enrichment workflows and methodologies. Due to differences in probe design, workflow, and synthesis, where probe selection algorithms can be private at the discretion of the manufacturer, it is hard to directly compare enrichment. Additionally, methods that may result in successful capture may be a result of multiple factors in the workflow, not just the probe design. Whilst probes may have a higher coverage of target nucleotides, this does not always correlate with more enrichment success. The O’Flaherty et al. (2018) [46] conserved panel had between 76% and 79% less probe coverage than the virus-specific panel; however, they only saw a 5.45% less overall increase in target reads by comparison (Figure 2). 

Capture rates are also highly variable within studies for many reasons. Varying virus titres in mixed infections, variance in concentration of pooled samples, and probe density across the genome can all contribute to widely varied enrichment for virus capture by any given method.

There are many variables that determine the success of enrichment. As each target was enriched to a different level, despite using the same probe design strategy, in any given study, there could be sequences that are highly enriched and others that are not enriched. Because of this, taking an enrichment average does not necessarily give an indication of how successful or poor a probe set may be at hybridising to a given series of targets. Additionally, as these large-scale probe sets aim to target a large and diverse range of viruses, they are not optimised for each virus individually. Another limitation is the terminology used in such studies. Terminology such as ‘coverage’ can often imply ‘depth’ as well as the number of nucleotides of the target organisms, with studies often categorising overlapping probes over a region as increased coverage as well as increased depth. This contributes to difficulty in assessing the coverage and depth of probe tiling for sequences. 

From the analysis of key virus hybrid-capture studies, enrichment success is not directly correlated to the methods used in the design of the probes. Despite a wide range of variables being present for large-scale virus panels, there was no clear indication of any one design consideration impacting the enrichment efficiency. There was also limited data to indicate whether design choices such as probe length and spacing also played an impact on the hybridisation potential. Workflows, such as the sequencing method used, are also factors that need to be investigated to fully understand the effectiveness of the capture. Additionally, the reason for enrichment, whether that be strain identification of generic sample enrichment prior to sequencing, may also impact enrichment effectiveness.

## 4. Hybrid-Capture in Planta

### 4.1. HybCap Enrichment of Plant Pathogens 

Target enrichment sequencing is a versatile tool that has had broad application in humans and vertebrates for pathogen identification and characterisation [36,37,44]. In plant hosts, there have been fewer applications, with HybCap enrichment for pathogen identification demonstrated in plant hosts to target bacteria [12] and fungi [14]. Additionally, it has also been used for population genetics of downy mildews (*Pseudoperonospora* sp.) [62] and identifying resistance genes in potatoes [63]. Plant diagnostics present many of the same limitations that are seen in humans. For example, PCR fails to provide enough resolution to differentiate strains of unculturable Candidatus liberibacter spp (CLas). Currently, the 16s rRNA gene is used to define CLas species; however, low copy numbers of pathogen DNA, differences in genome length between host and target, and a high pathogen titre are required for metagenomic sequencing. These challenges require alternative detection approaches to be taken for pathogen identification compared to vertebrate hosts. 

Cai et al. (2019) [12] adopted the Agilent SureSelect system for probe design to capture Candidatus liberibacter spp. Using 12,620x 120 bp RNA probes, Cai et al. (2019) [12] were able to obtain 99% coverage and 250X depth of coverage with enrichment compared to 65% coverage without enrichment. Despite this, 50% of reads were discarded due to inefficient read length and probe contamination. 

Probes have been designed and used for enrichment and sequencing with ONT to identify fungal pathogens [14]. The orthologs of targeted genes were extracted by Yu et al. (2023) [14] from 386 reference genomes of fungal species spanning six phyla to identify homologous regions that were used to design the baits used for enrichment. For probe design, DNA sequences were first clustered, and then consensus sequences within each cluster were identified to produce 26,000 probes that targeted 114 genes. To test the efficacy of the probe panel, the enrichment and sequencing of three species representing Ascomycota and Basidiomycota fungi were compared. The efficiency of enrichment, quantified as mean target coverage over the mean genome-wide coverage, ranged from 200 to 300. Furthermore, the enrichment of long reads increased the depth of coverage across the targeted genes and into non-coding flanking sequences. The assemblies generated from enriched samples provided well-resolved phylogenetic trees for taxonomic assignment and molecular identification.

Oomycetes are fungus-like microbes and contain plant pathogens belonging to the Phytophthora and Pythium genera. Many Oomycetes cannot easily be cultured in labs without their hosts [13]. Therefore, the quantity and quality of available samples for assay design and optimisation presents a problem. Additionally, Oomycetes are often embedded deep in host tissue, causing heavy contamination of exogenous host DNA. A study by Nguyen et al. (2021) [13] and Lemmon et al. (2012) [16] found that anchored hybrid enrichment was suitable for reducing the representation of both exogenous and endogenous non-target DNA [64]. Using 70 bp long probes with 3.5X tiling density, probes were designed to be highly specific, targeting only single-copy orthologous genes. Despite this, off-target fungi and plant contigs were still produced, likely due to the inclusion of probes targeting three barcoding COX genes.

### 4.2. Challenges Associated with Plant Virus Capture Panels

Genome composition, environmental factors, and selective pressures contribute to an ever-changing diversity in smaller virus populations as host–virus interactions are restricted to local communities [65,66,67]. If there is an increased scale to capture viruses with increasingly diverse genomes, the limited number of conserved elements and viral diversity complications for targeting universal genes need to be addressed. Specific genes have been found to be consistent with movement function or types of transmission, and these genes have been used for virus identification. These genes will be primary candidates for enrichment to capture viral genomes. Probe-based approaches are well placed to target diverse populations due to the probe’s ability to bind to divergent sequences and their scalable nature, and, when combined with sequencing, may overcome detection challenges. Whilst there has been considerable effort focusing on probe design considerations for human viruses, there is yet to be a commercial panel designed for the capture of plant virus targets. 

Different geographical areas have different priority target pathogens due to varying local commodities and threats to a region’s industries and native flora and fauna. For these panels to be commercially viable, they need to be large enough to account for multiple regions and industries, but targeting a diverse range of viruses increases the complexity of the panel. As a result, commercially available virus panels neglect priority plant viruses or are not optimised for clinical samples and thus are not suitable for plant diagnostics or use in a regulatory setting. Pecman et al. (2017) [56] demonstrated the use of ribosomal depleted RNA prior to sequencing of plant viruses to increase sensitivity. However, the identification of multiple diverse viruses using a single panel remains elusive.

## 5. Future Research Recommendations 

There is scope for target enrichment and HTS to solve some of the problems that have arisen with modern virus detection methodologies. Next-generation sequencing of all available nucleic acids in a sample may generate sufficient reads from pathogens to make an appropriate identification and may fail to generate sufficient reads when the pathogen titre is low. To increase sequencing resolution, hybrid capture methods have been used prior to sequencing to overcome this limitation. HybCap has been successfully used to enrich numerous pathogens in vertebrate species for characterisation and identification purposes [37,45]. Nevertheless, virus enrichment poses a challenge as viruses are highly diverse among families, and this has meant that additional design features need to be accounted for to ensure successful enrichment relative to fungal or bacterial pathogens. 

The mechanisms that allow for the large-scale enrichment and capture of nucleic acids are still relatively new. HTS is a tool that is growing in both popularity and application, and large-scale capture is slowly following. Confidence in panels could be improved by identifying more specific conditions that allow for probe binding as currently anywhere from 60% sequence identity can promote capture. Bead-washing to purify nucleic acids prior to capture can significantly reduce the number of target reads and lower the sensitivity of workflows. Automatic bead washing instruments are already increasing the consistency between experiments in this space, despite the cost. Additionally, with the rise of in silico capture work being conducted, the field would benefit from the testing of these algorithms on tissue samples to evaluate HybCap efficiency. These advancements may help to reduce the large ranges of enrichment efficiency that are currently seen with large panels. This would provide confidence so that large panels could be implemented into routine diagnostics. It is recommended that future research into this space be focused on simplifying HybCap workflows so that they are more accessible in order to reduce optimisation cost and time.

To summarise, in vertebrate virus enrichment studies, little correlation was found between probe coverage of target species and the enrichment that resulted. Studies demonstrated that by targeting conserved gene regions, enrichment was still successful despite using a fewer number of probes for a large number of gene targets [43,46]. Whilst there are many commercially available vertebrate virus panels for use with HybCap, plant viruses have been neglected. In plant hosts, large-scale HybCap panels were demonstrated to be successful in removing host rRNA using subtractive hybridisation to enrich viruses [56], and also enrichment of fungal genes prior to sequencing and identification [14]. There is no doubt that biosecurity would benefit from a large-scale screening and detection system for imported commodities. The question remains whether a large-scale hybridisation capture panel may be used to overcome modern diagnostic challenges for a diverse set of viruses in plant tissue. 

## Figures and Tables

**Figure 1 ijms-25-08348-f001:**
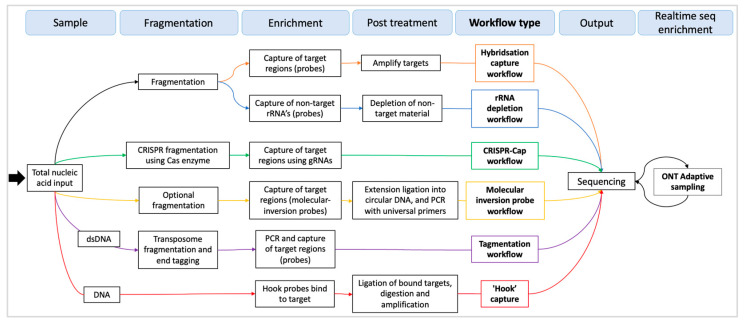
Diagram of HybCap and related enrichment workflows.

**Figure 2 ijms-25-08348-f002:**
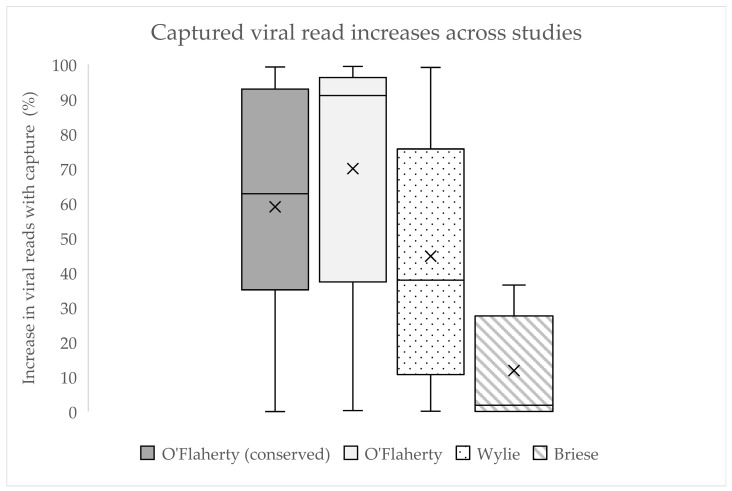
Box plot of increase in viral read capture in probe-based enrichment studies targeting viruses. “X” represents the mean. Values were calculated by calculating the percentage increase in viral reads in captured libraries compared to viral reads in uncaptured HTS libraries. For Wylie, this was calculated using the percentage of reference bases covered with captured and uncaptured HTS libraries [32,42,43]. Additional information can be found in Appendix A: Viral read Sup.

**Table 1 ijms-25-08348-t001:** Summary of Enrichment categories and types.

Enrichment Category	Enrichment Type	Example Method/Platform
Hybridisation capture	Array capture	Nimblegen capture array
	In-solution capture	Twist Bioscience capture, VirCapSeq-VERT, SureSelect
	Molecular inversion probes	smMIP
	Tagmentation method	TaME-seq, TaME-Seq2
	‘Hook’ capture	CircLigase enzyme
Subtractive hybridisation/background depletion	rRNA depletion	PolyA Enrichment, RiboZero
	Adaptive sampling	ONT adaptive sequencing
Amplicon-based enrichment	Multiplex PCR	Anchored multiplex PCR
		Rolling circle amplification
CRISPR-based enrichment	CRISPR enrichment	CRISPR-Cap

**Table 2 ijms-25-08348-t002:** Summary of related studies.

Paper	Study	Novel Platform	Library Preparation	Sequencing Platform	Probe Design	Targets	Enrichment of Target Reads
O’Flaherty et al. 2018 [46]	Virus characterisation using virus-specific probes and conserved probes	n/a	Tru-Seq RNA library	Illumina MiSeq	Custom panel, whole genome and protein coding sequence tiling for virus-specific probes. Conserved panel designed on viral groupings, consensus and degenerate sequences generated from algorithms	34 respiratory viruses (7 virus families) Conserved probes target 50 respiratory viruses (9 virus families)	7285-fold median increase (virus-specific probes) 8990-fold median increase (conserved probes)
Wylie et al. 2015 [32]	Virus characterisation using HybCap probes	ViroCap (probe design)	KAPA low throughput library construction kit on sonicated and pre-amplified libraries	Illumina HiSeq (2000 and 2500)	ViroCap, NimbleGen capture design	34 vertebrate virus families	296–674-fold median increase
Briese et al. 2015 [45]	Virus characterisation using HybCap probes.	VirCapSeq-VERT (Library prep)	VirCapSeq-VERT and conventional Illumina HTS	Illumina HiSeq (2500)	Vertebrate viruses clustered at 96% similarity, 100-mer oligos spaced by 25–50bp along sequences. Mutant sequences varying by 90% similarity kept	6 vertebrate virus families	100–10,000-fold increase
Li et al. 2020 [43]	Use of enrichment for cheaper and sensitive sequencing of Coronavirus genomes in bats	n/a	TruSeq stranded mRNA library preparation kit, Enrichment performed using xGen baits and modified NimbleGen workflow	Illumina HiSeq and Sanger sequencing	Custom panel synthesised using xGen lockdown baits	Coronavirus genomes (Bat)	10–1000-fold increase
Depledge et al. 2011 [44]	Virus characterisation using CBPH (custom panel) followed by Illumina sequencing	n/a	SureSelect target enrichment system	Illumina Genome Analyser IIx	120-mer RNA probes designed using custom Perl scripts and Agilent eArray software (https://earray.chem.agilent.com/earray/, accessed on 27 April 2023). Synthesised by Agilent	Herpesvirus	exceeded 100-fold read depth
Pecman et al. 2017 [56]	rRNA enrichment to assess subtractive hybridisation for Illumina sequencing of plant viruses	n/a	TailorMix miRNA Sample preparation kit v2 and Illumina HiSeq. Qiagen RNeasy Mini Kit and ScriptSeq complete kit (plant leaf) followed by MiSeq	Illumina HiSeq (2500), MiSeq	n/a	ssRNA virus and dsDNA virus	n/a
Cai et al. 2019 [12]	Enrichment for capture and characterisation of *Clavibacter liberibacter* strains	n/a	SureSelect target enrichment system, TruSeq PCR free DNA library preparation kit	Illumina MiSeq	120-mer RNA probes designed and synthesised by Agilent	*Candidatus Liberibacter asiaticus*	500–45,000-fold increase
Nguyen et al. 2021 [13]	Enrichment of Oomycete orthologues	n/a	Modified KAPA HyperPrep kit	Illumina MiSeq	Custom filtering of Oomycete sequences, 70-mer probes synthesised by myBaits	426 Oomycete orthologues	18-fold median increase
Yu et al. 2023 [14]	Fungal enrichment using HybCap enrichment and nanopore sequencing	n/a	TwistBioscience enrichment, ligation sequencing library preparation	Nanopore sequencing (MinION)	Twist custom panel	114 fungal genes spanning 6 phyla	200–300-fold increase

## Data Availability

The original contributions presented in the study are included in the article/Appendix A, further inquiries can be directed to the corresponding author.

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
