# Peer review of "A Review of Probe-Based Enrichment Methods to Inform Plant Virus Diagnostics"

_ijms, 2024, doi:10.3390/ijms25158348_

Round 1

Reviewer 1 Report

Comments and Suggestions for Authors

The authors present an interesting summary of the problem of viral plant pathogen detection with an emphasis on enrichment methods. The language of this manuscript is clear and understandable. However, I miss a summarizing scheme representing the comparison of alternative enrichment methods. The bibliography well describes the current state of the art on that subject. Therefore my recommendation for this review article would be that this manuscript deserves to be published in the International Journal Of Molecular Sciences after minor revisions.

Line 34 this sentence lacks a predicate

Line 51: Please also include ONT limitations

Line 168 Please summarize this paragraph in the form of a table enlisting strengths and weaknesses of presented methods, and information of the subject study. I understand that some aspects of these methods are hard to compare, therefore it is even more important to present the details of each method.  

I would suggest adding a scheme with alternative enrichment methods that can increase detection sensitivity. Try drafting the figure with the workflow for alternative methods for viral plant pathogens drawing attention to the alternative enrichment methods.

The comparison of the enrichment methods is a key element of this review. The comparison should be summarized in tabular and/or graphical form. Please try to ask yourselves who and how can benefit from your review. In my opinion, the scientists who would benefit the most from your review article would be people who try to create a new diagnostic method of viral plant pathogen detection. 

Author Response

Dear reviewer, 

Thank you for your comments. The authors acknowledge and thank you for your time preparing your feedback. In response to your comments:

Comment 1: Line 34 this sentence lacks a predicate

Response 1: We agree and have fixed sentence as suggested

Comment 2: Line 51: Please also include ONT limitations

Response 2: We agree and have included more information on ONT limitations

Comment 3: Line 168 Please summarize this paragraph in the form of a table enlisting strengths and weaknesses of presented methods, and information of the subject study. I understand that some aspects of these methods are hard to compare, therefore it is even more important to present the details of each method.  

Response 3: We agree and have summarised the details of sections from lines 168 into a table as suggested. The table gives better context for the categorisation of enrichment and where hybridisation capture techniques fit into the overall picture.

Comment 4: I would suggest adding a scheme with alternative enrichment methods that can increase detection sensitivity. Try drafting the figure with the workflow for alternative methods for viral plant pathogens drawing attention to the alternative enrichment methods.

Response 4: We have taken your feedback and included a figure highlighting alternative enrichment method workflows.

The figure aims at providing different detail on the HybCap workflows for diagnosticians that may be interested.

We have also improved readability across the document by making sentences more concise, introducing subheadings and removing non-relevant information. 

Once again we thank you for your input. 

Reviewer 2 Report

Comments and Suggestions for Authors

The review of Farrall and al. Entitled “A Review of Probe-based Enrichment Methods with Focus on Plant Virus Diagnostics” competently summarizes some of the most important progresses in the enrichment of plant pathogens, especially of viruses. The macrostructure of the review is OK, but the content could have been organized better in subsections rather than paragraphs that sometimes are spaced and sometimes are not. The single figure and table are not making the review very appealing. Some drawings outlining the workflow for some new technologies would be some quite nice additions.

Other comments

Line 161

Italicize Plasmodium falciparum

162-163

Italicize Borrelia  burgdorferi

Line 183-184

The following sentence has to be revised.

“The following studies detail attempts to design and capture viruses in a range of hosts using enrichment workflows.”

Why “the following studies detail”. The studies might provide details on the various methods but not this review.

To “design” what? How to “capture viruses”? Then why using “and”?

Table 1

It is not clear why the authors are listing the work of Cheng et al 2015 on Cancer genotyping using HybCap probes, Li et al 2020 on Use of enrichment for cheaper and sensitive sequencing of Corona-virus genomes in bats and so on. Is this a review related to plant technology or not? Were these technologies applied to enrich plant viruses?

Author Response

Dear reviewer, 

Thank you for your comments. The authors acknowledge and thank you for your time preparing your feedback. In response to your comments:

Comment 1: Line 161 Italicize Plasmodium falciparum.  

Response 1: We agree and have corrected this in the paper as suggested

Comment 2: 162-163 Italicize Borrelia  burgdorferi.

Response 2: We agree and have corrected this in the paper as suggested

Comment 3: Line 183-184, The following sentence has to be revised.

“The following studies detail attempts to design and capture viruses in a range of hosts using enrichment workflows.”

Why “the following studies detail”. The studies might provide details on the various methods but not this review.

To “design” what? How to “capture viruses”? Then why using “and”?

Response 3: We agree this is poorly worded and have corrected this in the paper as suggested

Comment 4: Table 1

It is not clear why the authors are listing the work of Cheng et al 2015 on Cancer genotyping using HybCap probes, Li et al 2020 on Use of enrichment for cheaper and sensitive sequencing of Corona-virus genomes in bats and so on. Is this a review related to plant technology or not? Were these technologies applied to enrich plant viruses?

Response 4: In table 1 (now Table 2) we have taken out some of the less relevant studies as suggested, however we have kept the studies we believe are relevant. We have kept vertebrate virus such as coronavirus enrichment in bats which we believe are useful for analysis of virus capture given the lack of studies looking at direct plant virus capture. Other studies such as hybrid capture conducted in plant matrices but not targeting viruses can still give context and comparison for capture from plant tissue. 

As suggested we have fixed the spacing and introduced subheadings to improve the readability. 

We have also improved readability across the document by making sentences more concise, removing irrelevant information and using more consistent terminology. 

Other changes include;

A new figure has been included that summarises some of the HybCap workflows, including new enrichment strategies, in the diagnostic space. 

A new table has also been added explaining the different enrichment types and some of the methods. 

The conclusion has been made more concise and now includes more information on suggested directions for future research. 

Once again we thank you for your input.

Reviewer 3 Report

Comments and Suggestions for Authors

The manuscript discusses various aspects of viral nucleic acid capture and enrichment techniques, focusing on hybrid capture methods and their application in plant virology. While the manuscript is comprehensive and informative, several issues need addressing to improve clarity, structure, and scientific rigor. It appears the manuscript was written in a hurry. The subject of the review is appropriate, but the authors should engage in more intensive discussions before drafting the review.

Major comments

Although the title suggests a focus on plant viruses, many parts of the manuscript deal with other pathogens. For example, references to vertebrate viruses should be removed from the abstract.

Table 1 contains numerous studies unrelated to plant viruses. Focus exclusively on plant viruses in the table.

Redundant points are repeated across different sections, particularly regarding the limitations of probe-based enrichment. Streamline the content to avoid repetition.

Some descriptions are vague or unclear. For example, the benefits of using RNA as an input need more specificity.

Simplify overly complex sentences for better readability and ensure consistent use of terminology.

Important technical details are sometimes missing, such as the conditions under which experiments were conducted. Ensure these details are included.

Provide clearer comparisons when discussing different studies. Include more quantitative data and direct comparisons of results.

Figure 1 does not look professional. Please prepare a new, more polished version.

Include more figures and tables to summarize data and improve readability. For instance, a table comparing the performance of different probe sets would be helpful.

Table 1 contains many unnecessary previous studies. Focus solely on plant viruses.

Add an abstract that summarizes the key points.

Make the conclusion more concise and clearly state the implications of the findings.

Discuss potential improvements in hybrid capture methodologies and future research directions.

Overall, I do not recommend the manuscript for publication in its current form. It would be more appropriate to submit the manuscript to a journal specializing in virology after rigorous revisions.

Comments on the Quality of English Language

Minor editing of English language required.

Author Response

Dear reviewer, 

Thank you for your comments. The authors acknowledge and thank you for your time preparing your feedback. In response to your comments:

Comment 1: Although the title suggests a focus on plant viruses, many parts of the manuscript deal with other pathogens. For example, references to vertebrate viruses should be removed from the abstract.

Response 1: Due to significant lack of plant studies using hybcap for large scale virus capture, there are little relevant studies to draw from, thus we feel that analysing vertebrate virus studies are an important part of assessing and comparing the design of probes. We also feel that discussing enrichment based diagnostics in plants targeting alternative pathogens is important and  informative. This paints a comprehensive picture at the current state of hybcap pathogen detection in plants. We do agree that the title could be more focused and have reworded the title better reflect the content of the paper. 

Comment 2: Table 1 contains numerous studies unrelated to plant viruses. Focus exclusively on plant viruses in the table.

Response 2: In table 1 (now table 2) we have taken out some of the less relevant studies as suggested, however we have kept the studies we think are relevant. We have kept vertebrate virus which we believe are useful for analysis of virus capture given the significant lack of studies looking at direct plant virus capture. Other studies, such as hybrid capture conducted in plant matrices, whilst not targeting viruses, still give context and comparison for capture from plant tissue. 

Comment 3: Redundant points are repeated across different sections, particularly regarding the limitations of probe-based enrichment. Streamline the content to avoid repetition.

Response 3: We agree and have reviewed the content of the manuscript. Revisions have been made to the language including taking out retreating points, such as that mentioned above. 

Comment 4: Some descriptions are vague or unclear. For example, the benefits of using RNA as an input need more specificity.

Response 4: In our manuscript revisions, we have gone into further detail of points that may not have been clear. The use of RNA has been given further context and clarity. 

Comment 5: Simplify overly complex sentences for better readability and ensure consistent use of terminology.

Response 5: We agree that some sentences may be unnecessarily complex and therefore have changed many complex sentences throughout the paper to improve the readability. We have also changed much terminology around enrichment and hybridisation capture so that it is consistent and easier to understand. 

Comment 6: Important technical details are sometimes missing, such as the conditions under which experiments were conducted. Ensure these details are included.

Response 6: When comparing studies, we have included all the information on conditions we believe to be relevant and influential to the results we are comparing, such as probe specifications, length etc. We have included more technical details of studies for further context of alternative enrichment methods.

Comment 7: Provide clearer comparisons when discussing different studies. Include more quantitative data and direct comparisons of results.

Response 7: We have made comparisons where we think appropriate, however the paper also discusses the limitations and challenges with drawing comparisons from such large and variable workflows. 

Comment 8: Figure 1 does not look professional. Please prepare a new, more polished version.

Response 8: We have prepared a new version of figure 1 (now figure 2) to look more professional and polished 

Comment 9: Include more figures and tables to summarize data and improve readability. For instance, a table comparing the performance of different probe sets would be helpful.

Response 9: We have added an additional table and figure, summarising the hybridisation workflows and characterisation of methods that have been seen in past papers and emerging technologies. Figure 2 currently compares the performance of probe sets from significant studies however there are limitations. Probe set comparisons are inherently hard to compare due to the large variability, context and purpose. This has been elaborated upon in the paper. 

Comment 10: Table 1 contains many unnecessary previous studies. Focus solely on plant viruses.

Response 10: See response 2

Comment 11: Add an abstract that summarizes the key points.

Response 11: We are slightly confused by this comment as an abstract is already present that details the aim of the review. Could you please elaborate. 

Comment 12: Make the conclusion more concise and clearly state the implications of the findings.

Response 12: We agree the conclusion can be improved to be more concise and targeted. We have removed the conclusion ‘section’ and reduced the conclusion to a final concluding paragraph. 

Comment 13: Discuss potential improvements in hybrid capture methodologies and future research directions.

Response 13: We agree this could be expanded upon more. We have changed section 5 (previously a conclusion), into a section summarising future direction of research and suggestions for the field. 

Significant changes and specific changes suggested by reviewers have been highlighted in the paper for reference.

Overall, the new figures and tables provide a more detailed perspective on the hybridisation enrichment workflows including comparison between notable HybCap studies. More thorough examination has been included explaining gaps in the literature and future directions. The readability and flow has been significantly improved across the whole of the text. 

Once again we thank you for your input.

Round 2

Reviewer 3 Report

Comments and Suggestions for Authors

As the authors mentioned, there were not enough publications associated with the given subject.

However, the authors did their best to revise the manuscript according to the reviewers' comments.

The newly included figure is helpful for readers to understand HybCap and related enrichment workflows.

I am quite satisfied with the revised manuscript.

The manuscript is suitable for publication as it is.

Congratulations on the excellent work!